# Chemoinformatic Screening for the Selection of Potential Senolytic Compounds from Natural Products

**DOI:** 10.3390/biom11030467

**Published:** 2021-03-22

**Authors:** Oscar Salvador Barrera-Vázquez, Juan Carlos Gómez-Verjan, Gil Alfonso Magos-Guerrero

**Affiliations:** 1Department of Pharmacology, School of Medicine, Universidad Nacional Autónoma de México (UNAM), Mexico City 04510, Mexico; osbarrera6@gmail.com; 2Dirección de Investigación, Instituto Nacional de Geriatría, Mexico City 10200, Mexico; jverjan@inger.gob.mx

**Keywords:** chemoinformatics, senescence, natural products, senolytics, database, aging

## Abstract

Cellular senescence is a cellular condition that involves significant changes in gene expression and the arrest of cell proliferation. Recently, it has been suggested in experimental models that the elimination of senescent cells with pharmacological methods delays, prevents, and improves multiple adverse outcomes related to age. In this sense, the so-called senoylitic compounds are a class of drugs that selectively eliminates senescent cells (SCs) and that could be used in order to delay such adverse outcomes. Interestingly, the first senolytic drug (navitoclax) was discovered by using chemoinformatic and network analyses. Thus, in the present study, we searched for novel senolytic compounds through the use of chemoinformatic tools (fingerprinting and network pharmacology) over different chemical databases (InflamNat and BIOFACQUIM) coming from natural products (NPs) that have proven to be quite remarkable for drug development. As a result of screening, we obtained three molecules (hinokitiol, preussomerin C, and tanshinone I) that could be considered senolytic compound candidates since they share similarities in structure with senolytic leads (tunicamycin, ginsenoside Rb1, ABT 737, rapamycin, navitoclax, timosaponin A-III, digoxin, roxithromycin, and azithromycin) and targets involved in senescence pathways with potential use in the treatment of age-related diseases.

## 1. Introduction

Cellular senescence is a biological condition that involves significant changes in gene expression and the loss of proliferative potential. Interestingly, during aging, immune clearance defects lead to the accumulation of senescent cells, which are thought to limit tissue by releasing the so-called senescence-associated secretory phenotype (SASP), thus creating a mild but chronic proinflammatory microenvironment that may impair the normal function of surrounding tissue [1]. Such an environment has been associated with several diseases such as cancer, cardiovascular disease, obesity, type 2 diabetes, sarcopenia, pulmonary fibrosis, osteoarthritis, atherosclerosis, and neurological disorders [1]. Moreover, senescent cells have upregulated pro-survival/antiapoptotic mechanisms such as the PI3K/Akt and Bcl2 pathways [2]. Studies in experimental models have shown that targeting senescent cells through different genetic or pharmacological methods may delay, prevent, and/or improve multiple adverse outcomes related to age [3].

Senolytics are a class of drugs that are distinguished by selectively killing senescent cells (SCs). The first reported compounds with senolytic properties were dasatinib, quercetin, fisetin, and navitoclax; however, over the years, this group of drugs has been slowly growing larger. It should be noted that in different preclinical models, treatment with senolytics allows the pro-survival Senescent Cell Anti-Apoptotic Pathways (SCAPs) to be temporarily deactivated, thereby causing the apoptosis of SCs in damaged tissues [4]. Moreover, in different preclinical studies, senolytics appear to restore progenitor dysfunction, attenuate tissue inflammation, and alleviate age-related metabolism and disease dysfunction in all cell and tissue types [5]. Senolytics have been shown to delay, prevent, or alleviate frailty, cancers, and cardiovascular, neuropsychiatric, liver, kidney, musculoskeletal, and pulmonary diseases and disorders, among others. The first pilot trials of senolytics suggest that they decrease senescent cells, reduce inflammation, and improve human frailty deficits [4,6]. Moreover, early trials with two senolytic compounds (quercetin and dasatinib) have demonstrated positive effects on clinical assessments of patients with idiopathic pulmonary fibrosis [7].

Natural products (NPs) have played an essential role in drug discovery; it is estimated that around 30% of the drugs currently used in clinics for the last 39 years are NP-derived or NP botanicals [8,9]. Additionally, different NPs have been shown to be quite helpful in therapeutics since they are known to possess exceptional selectivity for different cell targets such as quercetin [10]. Additionally, computational analyses have been very valuable for drug discovery and screening, improving the drug discovery success rate, and diminishing the number of experimental approaches needed. In this sense, the first senolytic drug (navitoclax) was discovered using chemoinformatic and network analyses [11]. Therefore, in the present work, we searched the literature for reported senolytic compounds in order to determine their physicochemical properties and use them, employing different methodologies based on fingerprinting and network pharmacology. The aim was to search for novel senolytic compounds in natural products across two databases: InflamNat (657 molecules from https://pubs.acs.org/doi/10.1021/acs.jcim.8b00560, Accessed on 8 June 2020) [12], and BIOFACQUIM, (422 molecules from https://biofacquim.herokuapp.com/). Accessed on 6 July 2020 [13].

## 2. Materials and Methods

### 2.1. Creation of a Senolytic Lead Dataset

#### 2.1.1. Collection of Senolytic Compounds

Review of the bibliography published in Pubmed and Scopus up to September 2020 concerning compounds reported as senolytic drugs to create a dataset.

#### 2.1.2. Selection of Senolytic Leads from the Dataset Based on Hierarchical Analysis and Chemical Space Analysis

The Simplified Molecule Input Line Entry System (SMILES) of the compounds from the dataset was searched in Pubchem, accessed on 3 August 2018 to 30 September 2020 (https://pubchem.ncbi.nlm.nih.gov/) [14] and the SwissADME server, accessed on 30 September 2020 (http://www.swissadme.ch) [15]. Osiris Data Warrior software V5.2.1 was used in order to analyze the physicochemical properties and molecular descriptors such as G protein-coupled receptors (GPCR) ligand (GPCR. Ligand), ion channel modulator (ion.channel.modulator), kinase inhibitor (kinase.inhibitor), nuclear receptor ligand (nuclear.receptor.ligand), protease inhibitor (protease.inhibitor), enzyme inhibitor (enzyme.inhibitor), number of violations (nviolations), number of atoms (natoms), log k p values in [cm/s] (log.Kp..cm.s.), Lipinski’s Rule violations (Lipinski..violations), Ghose Filter violations (Ghose..violations), Veber Rule violations (Veber..violations), Egan Rule violations (Egan..violations), Muegge’s Rule violations (Muegge..violations), bioavailability score (bioavailability.score), molecular weight (molweight), P: conc (octanol)/conc (water) (cLogP), S: water solubility in mol/L (cLogS), hydrogen acceptor (H.acceptors), hydrogen donors (H.donors), total surface area (total.surface.area), polar surface area (polar.surface.area), druglikeness (druglikeness), shape index (shape.index), molecular flexibility (molecular.flexibility), electronegative atoms (electronegative.atoms), rotatable bonds (rotatable.bonds), aromatic rings (aromatic.rings), aromatic atoms (aromatic.atoms), sp3 atoms (sp3 atoms), and symmetric atoms (symmetric.atoms) of the compounds previously obtained from the created dataset [16].

First, we performed a cluster analysis of all physicochemical and molecular descriptors using the K-means algorithm and a distance matrix from senolytic compounds. Such analysis was performed with the complexheatmap package [17] with R-studio (RStudio PBC250, Boston, MA, USA), version 3.4. Next, the chemical space and the distribution of the data were analyzed using Principal Component Analysis (PCA) to evaluate whether any of the parameters (molecular descriptors) were more critical in distributing the chemical space on the cluster analysis. PCA is defined as an orthogonal linear transformation technique that can transform the data into a new coordinate system. We used a two-dimensional system in our analysis [18]. Briefly, the variance of the data maximized on the first coordinate was called the first principal component. The second variance was maximized on the second coordinate; we used the factoextra package with R-Studio version 3.4.

### 2.2. Selection of Drug-Likeness of NPs from the BIOFACQUIM Database and the InflamNat Dataset Based on the Quantitative Estimate of Drug-Likeness (QED)

#### 2.2.1. Data Collection for NPs

We obtained a drug-likeness estimator based on the Quantitative Estimate of Druglikness (QED) for compounds from the InflamNat and BIOFACQUIM databases [12,13] by using DruLito software (http://www.niper.gov.in/pi_dev_tools/DruLiToWeb/DruLiTo_index.html), accessed on 7 December 2020 [19]. The Spatial Data File (SDF) files required for this software were obtained from the PubChem (https://pubchem.ncbi.nlm.nih.gov/) [14] and ZINC15 [20] servers, accessed on 11 January 2021. We analyzed only those molecules available on both servers.

#### 2.2.2. Determination of the Drug-Likeness of NPs

To determine drug-likeness, we used the Quantitative Estimate of Drug-likeness (QED), an integrated score, to evaluate the compounds’ preference to be considered a hit. QED is a method for the quantification of drug-likeness, considering the main molecular properties together. The QED score uses the molecular descriptors: molecular weight (Da), Ghose-Crippen-Viswanadhan octanol-water partition coefficient (AlogP), the number of H-acceptors (HBA), the number of H-donors (HBD), the number of rotatable bonds, total polar surface area (TPSA), and aromatic bond count. A histogram and boxplot were performed to compare the distribution of the molecular descriptors, both senolytic compounds and NP properties. We selected the NPs with high QED scores (over 0.5).

### 2.3. The Selection of Putative Senolytic Compounds Is Based on a Comparison of the Fingerprints of the Drug-Like NPs and Senolytic Leads

#### Fingerprints of Drug-Like NPs and Senolytic Leads

To determine their fingerprints and perform a comparison, we used ChemmineR and rcdk in R-Studio version 3.4 [21]. We used the extended value with a default length of 1024 (number of bits), taking rings and atomic properties of the senolytic compound dataset as leads and the NPs obtained from databases as tested compounds. Then, we performed a cluster analysis by using the Tanimoto coefficient [21] to compare the drug-likeness of NPs versus the senolytic compounds. The molecules were then classified into three groups using as a parameter the molecular distances by Ward’s clustering method, which are the most popular hierarchical clustering algorithms used in drug discovery [22]. To confirm the clustering results, the Dunn index and the silhouette coefficient were used [21]. After the different clustering methods were performed, the cluster containing the senolytic lead was analyzed in order to obtain the putative senolytic molecules and construct a compound-target network.

### 2.4. Selection of the Best Putative Senolytic Molecules through Their Multitarget Capacity

#### Compound-Target Network Generation

To explore the possible interaction of the drug-like molecules with targets involved in the senescence processes, we generated a compound-target network (by using Compound Spring Embedder (CoSE) layout based on the Cytoscape software, (National Institute of General Medical Sciences of the National Institutes of Health, Bethesda, MD, USA), 3.8.2 version [23]) with information on targets obtained from the BindingDB database (https://www.bindingdb.org/bind/index.jsp) [24], accessed on 12 January 2021. This database of measured binding affinities focuses on the interactions of proteins considered to be drug targets with small drug-like molecules [24]. Interestingly, this database has been previously used in search of anti-inflammatory drugs from NPs [18]. The network was constructed from several pathways involved in senescence present in Wikipathways such as DNA damage/telomere stress-induced senescence (DD/TSIS), pentose phosphate pathway and glycolysis in senescent cells (PPPsc), oncogene-induced senescence (OIS), DNA damage response (DDR), oxidative stress-induced senescence (OSIS), glycolysis in senescence (Gis), senescence-associated secretory phenotype (SASP), intrinsic pathways for apoptosis (IPFA) and the cell cycle (CC) [25,26] and information on BindingDB with compounds to select the multitarget putative senolytic molecules. For more details concerning the data on targets and respective pharmacological functions and the genes from the senolytic network, see the Appendix A. The methodology of this work is summarized in Figure 1.

## 3. Results

### 3.1. Creation of a Senolytic Lead Dataset

The search for senolytic compounds in Scopus and Pubmed showed 79 compounds with senolytic activity. Hierarchical clustering and PCA analysis performed with molecular descriptors can be seen in Figure 2 and Appendix A. Interestingly, we found two main clusters of senolytic compounds according to their 1-D and 2-D molecular descriptors focused on molecular.flexibility, H.Donors, Veber..violations, Muegge..violations, Lipinski..violations, nviolations, H.Acceptors, polar.surface.area, electronegative.atoms, sp3.atoms, molweight, natoms, total.surface.area, rotatable.bonds, Egan..violations, and Ghose..violations. The cluster with the greatest similarity between the molecules was cluster one (marked in yellow in Figure 2). This cluster includes navitoclax, ABT-737, tunicamycin A, B, C, and D, digoxin, rapamycin, ginsenoside Rb1, roxithromycin, timosaponin A-III, and azithromycin. We used these compounds as senolytic leads for the rest of the study. Table 1 summarizes the main characteristics of the compounds in cluster one, including their pharmacological activity.

### 3.2. Selection of Drug-Like NPs Based on QED

From 1079 NP molecules obtained from the InflamNat (657 molecules) and BIOFACQUIM (422 molecules) datasets, we only used 562 molecules analyzed in DruLito software to obtain the physicochemical parameters required for QED. The comparison of the physicochemical parameters and the whole senolyitic dataset (79 molecules) is shown in Figure 3 and Figure 4. In both figures, the molecular descriptor values of the NPs and senolytic compounds are distributed within a similar range.

After QED, molecules with a score greater than 0.5 were considered drug-like, producing only 345 molecules (61.39%) of the 562 NPs, from which we obtained Total Number of NP Molecules that Violated the Rule (weighted Quantitative Estimate of Druglikeness (wQED) < 0.5) = 217 and Total Number of NP Molecules that Passed the Rule (wQED > 0.5) = 345.

### 3.3. Selection of Putative Senolytic Compounds

From 345 drug-like molecules obtained by QED, only 53 molecules were considered in BindingDB; they showed reported targets. We performed a comparison of the 53 molecules with each of the 17 senolytic leads obtained from cluster one (Figure 2) through fingerprints, the Tanimoto coefficient, and Ward’s method. Figure 5 is an example of this analysis with rapamycin. The optimal number of clusters was determined by the Elbow method (Figure 5A), resulting in three clusters as an optimal number for cluster analysis. The Dunn index and the silhouette coefficient (Sc) corroborate the cluster analysis for the 53 drug-like molecules and the senolytic leads from group 1 (Figure 5D). The number of drug-like molecules clustered with senolytic leads from group 1 is 17 for navitoclax (Sc: 0.23), 16 for ABT 737 (Sc: 0.23), 17 for tunicamycin (Sc: 0.24), 14 for ginsenoside (Sc: 0.29), 16 for azithromycin (Sc: 0.27) and roxithromycin (Sc: 0.27), 14 for timosaponin A-III (Sc: 0.29), 11 for digoxin (Sc: 0.39), and 11 for rapamycin (Sc: 0.38). The mean of Sc was highest in three clusters, reflecting a better similarity between the drug-like molecules and the senolytic leads (Figure 5D). Appendix A shows the complete analysis for each of the 17 senolytic leads from group 1. Finally, the mean of the Sc of the cluster analysis performed with the 17 senolytic leads from group 1 was 0.27. Table 2 shows the results after eliminating repeated molecules.

### 3.4. Network Analysis to Obtain the Best Putative Senolytic Molecules through Their Multitarget Capacity

We obtained a compound-target network (Figure 6) related to senescence that shows 861 nodes and 1227 edges (or links), where 17 nodes represent the drug-like molecules obtained by QED analysis. It should be noted that three molecules have the most significant number of targets involved in any of the senescence pathways used. These putative senolytic molecules are represented in Figure 6 as yellow nodes and their cellular targets as red nodes. The putative senolytic molecule hinokitiol showed a connection to the senescence network through the inhibition of histone deacetylase 4 and 5 (HDCA4 and 5) [57], targets involved in the cell cycle (CC). Preussomerin C inhibits JUN protein, a target involved in oxidative stress-induced senescence (OSIS) [58], and tanshinone I modulates the target RAD51, which is involved in DNA damage response (DDR) [59] (Figure 6). Table 3 summarizes the main characteristics of these putative senolytic compounds obtained from this analysis.

The putative senolytic molecules with the most significant number of targets were hinokitiol, preussomerin C, and tanshinone. Hinokitiol showed a connection to the senescence network through the targets; hinokitiol inhibits histone deacetylase 4 (HDCA4) and 5 [58], which are involved in the cell cycle (CC). Preussomerin C showed a connection to oxidative stress-induced senescence (OSIS) through the target JUN, acting as an inhibitor [59], and tanshinone I was linked to DNA damage response (DDR) through the target RAD51 as a modulator [59] (Figure 6). Hinokitiol is a monoterpenoid proven to be helpful as an antimicrobial, antifungal, antiviral, antiproliferative, anti-inflammatory, and antiplasmodial agent [60,61]. It can be isolated from the roots of the Hinoki tree, *Hiba arboruitae* (Japanese cypress) [62,63]. Preussomerin C belongs to the Preussomerins family and exhibits a wide range of bioactivities, such as antibacterial, antifungal, nematicidal, and cytotoxic activity [63,64,65,66]. Finally, tanshinones are the main bioactive compounds from *Salvia miltiorrhiza* Bunge (Danshen) roots. They are extensively used in traditional Chinese medicine and have been reported to inhibit migration, invasion, and gelatinase activity in lung cancer cell line CL1-5 [53]. Table 3 summarizes some characteristics of the three putative senolytic compounds obtained from the compound-target network.

## 4. Discussion

Drug discovery based on natural products has a long and successful history. Interestingly, when natural product research is combined with computer-aided drug design techniques, an actual amount of success becomes evident, mainly when we explore natural products’ chemical spaces in order to identify bioactive compounds, emphasizing drug discovery [67]. In this context, the first senolytics were discovered using bioinformatics approaches to identify agents that transiently disable the SCAP networks that allow senescent cells to survive the hostile microenvironment they create and kill the non-senescent cells around them [4].

In this study, we identified 79 compounds in Pubmed and Scopus reported as senolytics, and once we performed the cluster analysis, we found two groups with similarities in their molecular descriptors. The most representative cluster, by similarities, included the following drugs: navitoclax, ABT-737, ginsenoside Rb1, tunicamycin A, B, C and D, timosaponin AIII, digoxin, rapamycin, azithromycin, and roxithromycin. These drugs are involved in cellular senescence [2,25,26]. This cluster was corroborated with PCA analysis. It is important to mention that all of them have different senolytic action mechanisms. For instance, navitoclax and ABT-737 can inhibit Bcl-2 and Bcl-xL, triggering the activation of caspases and exposure to phosphatidylserine in a Bak/Bax-dependent manner [68], and they share a similar sulfonylbenzamide structure. Tunicamycin binds irreversibly to the translocase enzyme [69]. Ginsenoside Rb1 is a glycosylated triterpenoid saponin that affects the Wnt/β-catenin signalling pathway by downregulating β-catenin/T-cell factor-dependent transcription and the expression of its target genes (ATP-binding cassette G2 and P-glycoprotein) [30,70]. Macrolides (azithromycin and roxithromycin) are inhibitors of protein synthesis via binding to the 50S ribosomal subunit of bacteria at the peptidyl transferase center, which enhances autophagosome formation of T cells by suppressing S6RP phosphorylation, thereby protecting against chronic inflammation [31,32,71,72]. Timosaponin AIII is a steroidal saponin with a selective cytotoxic activity that involves the inhibition of mTOR, the induction of ER stress, and protective autophagy [34]. Digoxin is a cardiac glycoside with a steroid-like structure used in order to treat cardiac arrhythmias and congestive heart failure [73], and rapamycin, a multitarget drug, is an inhibitor of mTOR complex 1 (mTORC1), which phosphorylates substrates including S6 kinase 1 (S6K1), eIF4E-binding protein 1 (4E-BP1), transcription factor EB (TFEB), unc-51-like autophagy-activating kinase 1 (Ulk1), and growth factor receptor-bound protein 10 (GRB-10). These substrates are involved in several processes including autophagy, cancer, neurological diseases, and aging [19,35].

To evaluate whether a compound is drug-like, Lipinski’s rule of five is traditionally used. However, the rule does not apply to substrates of biological transporters or natural products [74] which do not necessarily fall into Lipinski’s rule, and many approved drugs are a clear example of that (16% of oral drugs violate at least one of the criteria and 6% fail two or more) [74]. For this reason, Lipinski’s rule of five was replaced by the Quantitative Estimate of Drug-likeness (QED) as criteria for evaluating the drug-likeness of an NP [74]. This estimation allowed us to refine our search for NPs that could be candidates for drugs. In this context, the selection criteria were more tolerant in the QED, despite the fact that with other criteria, the molecular properties of some NPs could be outside the selection parameters, as occurs with Lipinski’s rule. QED allowed us to avoid bias and enrich our selection of NPs, obtaining a more realistic and graduated result. In reality, many approved drugs violate the criteria of this rule [74]. The distribution of the molecular descriptors between both datasets was similar, even though the senolytic dataset contained molecules of different natures, ranging from natural to synthetic products. Thus, these data indicate that natural products can be used as a reliable source for drug discovery.

Although there are many clustering methods, only some are used in practice; two were used in this study: hierarchical grouping and the K-means method. They have been used successfully in fingerprint analysis [21]. In this study, fingerprint analysis allowed us to obtain 17 molecules due to their similarity with lead molecules from cluster one, indicating that the physicochemical properties of the 17 compounds have a greater similarity with the properties of senolytics. Different clustering methods validated these analyses. The construction of a compound-target network helped us to select which compounds might be better senolytic candidates due to their therapeutic targets. All these data helped us to verify that the similarity in chemical structure may exhibit a similar function at the biological level, avoiding high costs in experiments, because some NPs (hinokitiol, preussomerin C, and tanshinone I) showed targets involved in some pathways related to senescence such as the cell cycle, oxidative stress-induced senescence (OSIS), and DNA damage response (DDR), which are linked to several conditions, disorders, and age-related diseases [4,75].

It ought to be noted, however, that these three remaining molecules may still need experimental validation to demonstrate their properties as senolytics. We should not overlook that many of these drugs’ targets have not yet been fully explored, so many putative senolytics obtained from fingerprint analysis may also show senolytic activity due to their structural similarity. However, the criterion we used in our work was to consider those NPs that showed similarity with senolytic leads and had targets reported and involved in senescence. This study helped us to identify which compounds of those previously selected by fingerprints possess multitarget capacity, in agreement with those proposed by Kirkland et al. (2020), who indicate that a senolytic is a compound that can interact with different nodes in different ways, increasing the specificity for senescent cells [4]. The present study identified three compounds by evaluating the multitarget structure and functionality, which increases the possibility of selective action on senescent cells. These new molecules open a field of pre-clinical study of great interest, which is why the use of computational tools is necessary to expand knowledge through the discovery of new candidates from NPs, and their evaluation in pre-clinical studies.

## 5. Conclusions

Our results show that the compounds hinokitiol, preussomerin C, and tanshinone-I can be used as senolytics due to their multitarget potential and their chemical characteristics of similarity. Therefore, they may be used in medicine to delay some age-related conditions such as cancer or neurodegenerative diseases. Nevertheless, further experimental analysis must be performed to validate such activities and demonstrate effectiveness and low toxicity. These agents may lead to the delay, prevention, or treatment of senescence and age-related conditions in humans, if clinical trials continue. Additionally, compounds from cluster one (tunicamycin, ginsenoside Rb1, ABT 737, rapamycin, navitoclax, rimosaponin A-III, digoxin, roxithromycin, and azithromycin) may be used as lead compounds for future chemoinformatic analysis due to the similarity in their molecular descriptors.

## Figures and Tables

**Figure 1 biomolecules-11-00467-f001:**
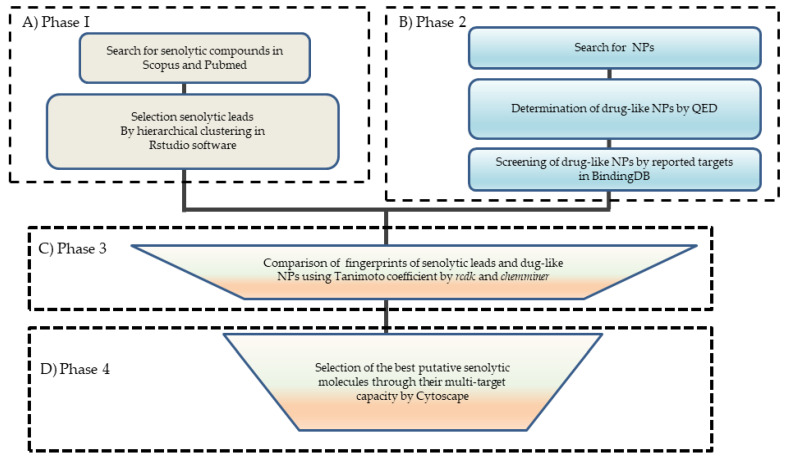
The workflow of the methodology for identifying putative senolytic molecules. (**A**) Phase 1: Creating a senolytic lead dataset from Pubmed and Scopus databases from the information obtained up to September 2020. The leads were determined by hierarchical structural clustering using the complexheatmap (K-means) with R-Studio through their molecular descriptors. (**B**) Phase 2: Selection of drug-like natural products (NPs) from the BIOFACQUIM and Inflamnat datasets based on the Quantitative Estimate of Drug-likeness (QED). The molecules were searched in the Pubchem server to obtain their structural information. Data filtering of NPs to obtain drug-like molecules by QED was searched for the reported targets in BindingDB. (**C**) Phase 3: Selection of putative senolytic compounds based on a comparison of the fingerprints of the drug-like NPs and senolytic leads employing the Tanimoto coefficient and the Silhouette clustering (used to corroborate the drug-like molecules from NPs). (**D**) Phase 4: Selection of the best putative senolytic molecules by creating networks with Compound Spring Embedder (CoSE) layout using Cytoscape software. The selected drug-like molecules, their pharmacological targets, and the network that contained targets involved in senescence were used in order to construct a compound-target senescence network.

**Figure 2 biomolecules-11-00467-f002:**
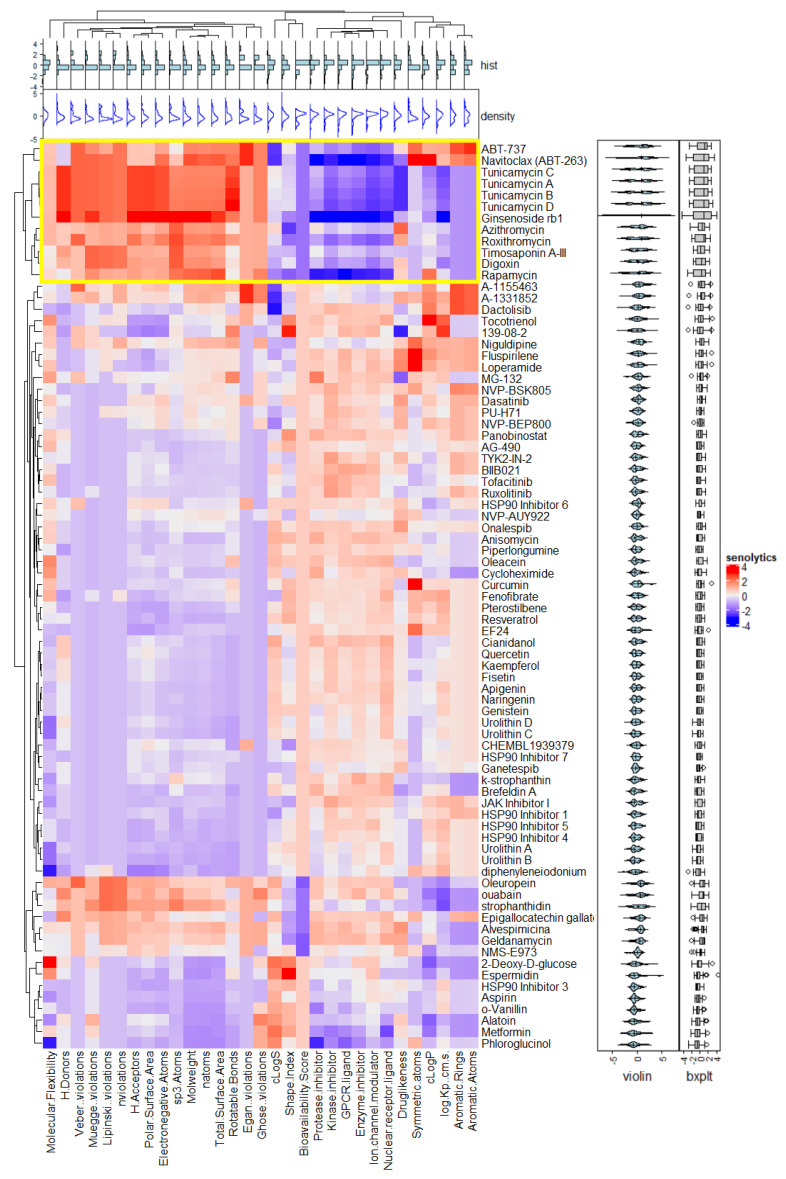
Hierarchical structural clustering of the previously reported senolytic compounds. A heatmap of hierarchical clustering was generated employing the complexheatmap package in R-Studio separated by K-means. The histogram at the top represents a frequency distribution of each molecular descriptor (turquoise). Density plots at the top represent the distribution of each molecular descriptor (blue). Violin plots on the right show the molecular descriptors’ distribution for each senolytic compound and their probability density. Boxplots on the right show the distribution of numerical data from molecular descriptors in each senolytic compound. As seen in the dendrogram at the left of the figure, there are two main clusters; we focus on the first, framed in red lines due to its similarity coefficient.

**Figure 3 biomolecules-11-00467-f003:**
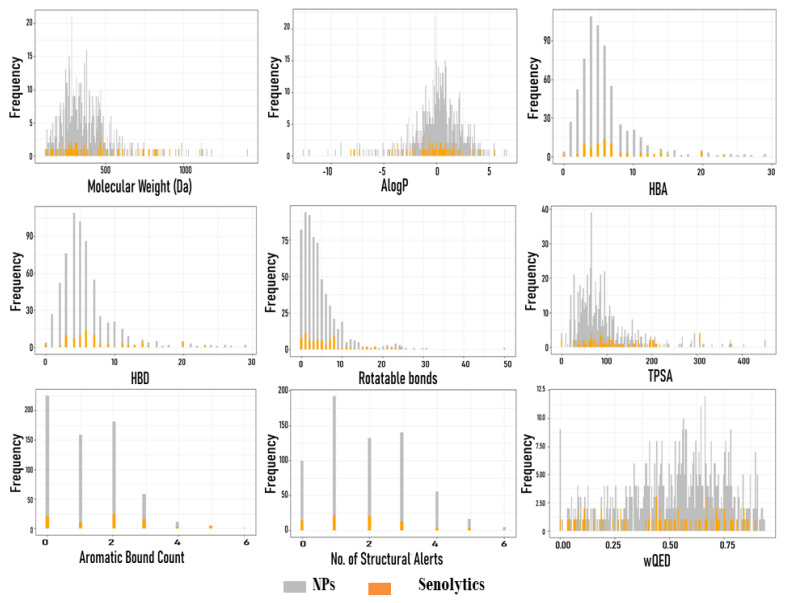
Distribution of molecular descriptors for senolytic compounds and NPs: molecular weight (Da), Ghose-Crippen-Viswanadhan octanol-water partition coefficient (AlogP), the number of H-acceptors (HBA), the number of H-donors (HBD), the number of rotatable bonds, total polar surface area (TPSA), aromatic bond count, and Quantitative Estimate of Drug-likeness (wQED). Vertical lines represent frequency values, while horizontal lines represent the intervals of the data. The abscissa axis represents the calculated values for each compound and each molecular descriptor, and the ordinate axis represents the frequency of the values. NPs (*n* = 562) and senolytic compounds (*n* = 79).

**Figure 4 biomolecules-11-00467-f004:**
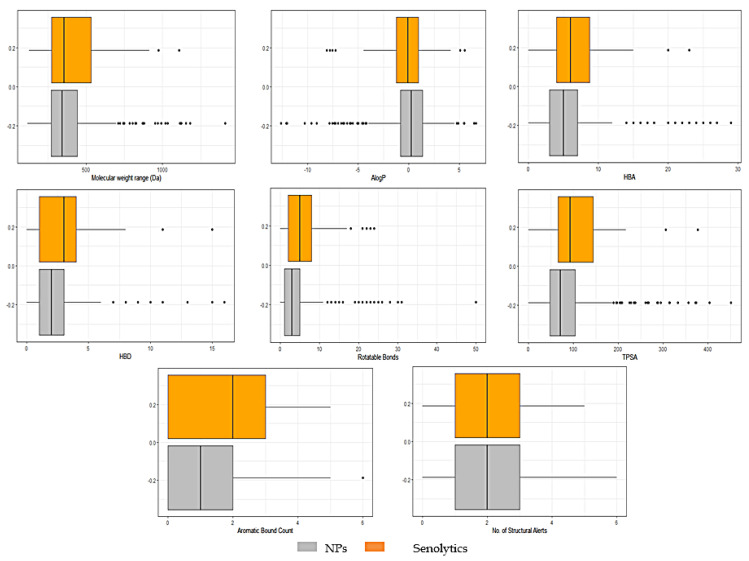
Boxplots showing the distribution and summary statistics of the molecular descriptors of NPs and senolytic compounds: molecular weight range (Da), Ghose-Crippen-Viswanadhan octanol-water partition coefficient (AlogP), the number of H-acceptors (HBA), the number of H-donors (HBD), the number of rotatable bonds, total polar surface area (TPSA), aromatic bond count of senolytic compounds and NPs. The abscissa axis represents the calculated values for each compound and each molecular descriptor, and the ordinate axis represents the frequency of the values. NPs (*n* = 562) and senolytic compounds (*n* = 79).

**Figure 5 biomolecules-11-00467-f005:**
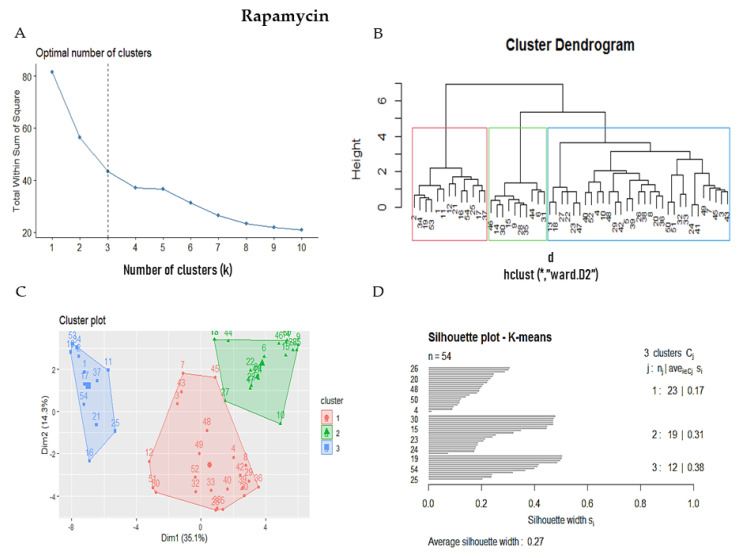
Steps in the selection of putative senolytics comparing 53 drug-like NPs with the senolytic lead rapamycin. (**A**) The dashed line represents the suitable number of clusters determined by the Elbow method. (***B***) Cluster analysis of Ward’s method, the senolytic cluster is shaped in red and the senolytic molecule as 1. (**C**) Cluster plot using Kmeans showed the same molecules in the senolytic cluster marked in blue. (**D**) Silhouette cluster representation to corroborate the previously described cluster analysis methods. * means matrix of distances used in clustering by Ward’s method in subfigure B.

**Figure 6 biomolecules-11-00467-f006:**
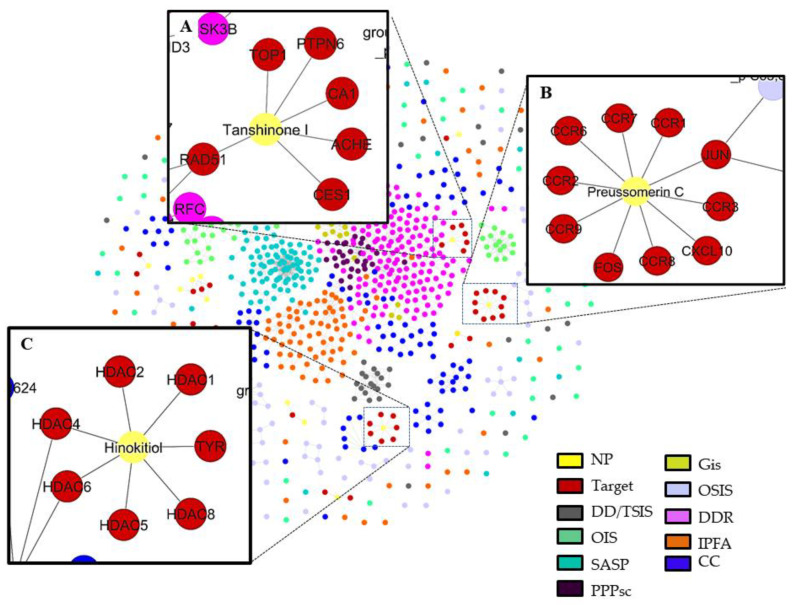
A network of putative senolytic compounds with their biological targets. (**A**) Interactions of tanshinone I with its phamarcological targets: TOP1, CA1,CES1, ACHE, PTPN6 and RAD51. (**B**) Interactions of preussomerin C with its phamarcological targets: CCR1,CCR2,CCR3,CCR6,CCR7,CCR8, CCR9, CXCL10, FOS and JUN. (**C**) Interactions of hinokitiol with its phamarcological targets: HDAC1, HDAC2, HDAC4, HDAC5,HDAC6, HDAC8, and TYR. The compound-target network was generated with CoSE Layout in Cytoscape software, version 3.8.2 [23]. The network was constructed with several pathways involved in senescence represented with the nodes of different colors and compounds (Methods): Nodes represent compounds or targets related to senescence according to their color. An edge represents a relation obtained from BindingDB. Abbreviations: DNA topoisomerase 1 (TOP1), carbonic anhydrase 1 (CA1), liver carboxylesterase 1 (CES1), acetylcholinesterase (ACHE), tyrosine-protein phosphatase non-receptor type 6 (PTPN6), DNA repair protein RAD51 homolog 1 (RAD51), CC chemokine receptor 1–3,6–9 (CCR1–3, 6–9), C-X-C motif chemokine ligand 10 (CXCL10), Jun Proto-Oncogene, AP-1 Transcription Factor Subunit (JUN), Fos Proto-Oncogene, AP-1 Transcription Factor Subunit (FOS), histone deacetylase 1,2,4–6,8 (HDAC1,2,4–6,8), tyrosine-protein phosphatase non-receptor type 6 (TYR), DNA damage/telomere stress-induced senescence (DD/TSIS), pentose phosphate pathway and glycolysis in senescent cells (PPPsc), oncogene-induced senescence (OIS), DNA damage response (DDR), oxidative stress-induced senescence (OSIS), glycolysis in senescence (Gis), senescence-associated secretory phenotype (SASP), intrinsic pathways for apoptosis (IPFA) and the cell cycle (CC).

**Table 1 biomolecules-11-00467-t001:** Main characteristics of the molecules of cluster one.

Senolytics	Pubchem Compound ID(CID)	Pharmacological Activity	K-Means Coefficient	References
Navitoclax	24978538	Inhibitor of Bcl-2 and Bcl-xL	0.44	[11,27]
ABT 737	11228183	Inhibitor of Bcl-2 and Bcl-xL	0.58	[28]
Tunicamycin	56927848	Disturbs the endoplasmic reticulum (ER) homeostasis and causes the accumulation of misfolded or unfolded proteins in the ER, inducing cell death	0.34	[29]
Ginsenoside Rb1	9898279	Affects the Wnt/β-catenin signalling pathway by downregulating β-catenin/T-cell factor-dependent transcription and expression of its target genes ATP-binding cassette G2 and P-glycoprotein	0.31	[30]
Azithromycin	447043	Enhances autophagosome formation of T cells by suppressing S6RP phosphorylation, which is a downstream target of the mammalian target of rapamycin (mTOR) pathway	0.32	[31]
Roxithromycin	6915744	Inhibitor of TGF-β1-induced activation of ERK and AKT and down-regulation of caveolin-1	0.23	[32]
Timosaponin A-III	15953793	The inductor of selective cytotoxic activity that involves inhibition of mTOR, induction of ER stress, and protective autophagy	0.29	[33]
Digoxin	2724385	Positive inotropic and negative chronotropic agent	0.21	[34]
Rapamycin	5284616	Inhibitor of mTOR complex 1 (mTORC1), which phosphorylates substrates including S6 kinase 1 (S6K1), eIF4E-binding protein 1 (4E-BP1), transcription factor EB (TFEB), unc-51-like autophagy-activating kinase 1 (Ulk1), and growth factor receptor-bound protein 10 (GRB-10)	−0.2	[35]

**Table 2 biomolecules-11-00467-t002:** Drug-like molecules obtained by fingerprint analysis and their pharmacological activity.

Compound Resulting from Fingerprint Analysis	ZINC ID	Pharmacological Activity Reported	References
Cacospongionolide B	ZINC26966472	Anti-inflammatory agent	[36]
Carnosol	ZINC3871891	Antineoplastic agent	[37]
Dihydrotanshinone I	ZINC2585546	Antiviral, anti-mutagenic, anti-cancer agent	[38,39,40]
Epoxyazadiradione	ZINC58576553	Anti-inflammatory agent	[41]
Farnesiferol B	ZINC29134693	Anti-oxidant agent	[42]
Friedelin	ZINC4097720	Anti-inflammatory and antipyretic agent	[43]
Fuscoside B	ZINC72123265	Anti-inflammatory agent	[44,45]
Gibberellic acid	ZINC3860467	Anti-inflammatory agent	[46]
Gliotoxin	ZINC3875454	Anti-inflammatory agent	[47]
Hinokitiol	ZINC95911093	Anti-cancer agent	[48]
Neurolenin B	ZINC100090140	Anti-inflammatory agent	[49]
Penicillic acid	ZINC3874657	Antibiotic	[50]
Preussomerin C	ZINC34383300	Cytotoxic and anti-nematodal agent	[51,52]
Tanshinone I	ZINC2558154	Anti-oxidant and anti-inflammatory agent	[40,53]
Tanshinone IIA	ZINC1650576	Anti-oxidant and anti-inflammatory agent	[54]
Triptolide	ZINC6483512	Anti-cancer, anti-inflammation, anti-obesity, and anti-diabetic	[55]
Ursolic acid	ZINC31356858	Anti-inflammatory and antihyperlipidemic agent	[56]

**Table 3 biomolecules-11-00467-t003:** Main characteristics of the best putative senolytic compounds obtained by multitarget capacity.

Senolytic Candidate	Structure	Source	Pharmacological Activity	Targets of Senolytic Compound Network	References
Hinokitiol ZINC95911093	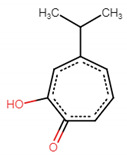	The roots of the Hinoki tree, *Hiba arboruitae* (Japanese cypress).	Anti-cancer agent	7	[48]
Preussomerin C ZINC34383300	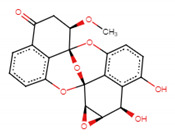	Endophytic fungus *Lasiodiplodia theobromae* ZJ-HQ1	Cytotoxic and anti-nematodal agent	10	[51,52]
Tanshinone I ZINC2558154	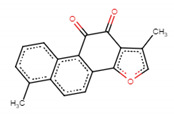	*Salvia miltiorrhiza* (Danshen or Tanshen in Chinese)	Anti-inflammatory, anti-coagulant, and anti-neoplasic agent	6	[40,53]

## Data Availability

The data that support the findings of this study are available in the Appendix A of this article.

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
