# Peer review of "Chemoinformatic Screening for the Selection of Potential Senolytic Compounds from Natural Products"

_biomolecules, 2021, doi:10.3390/biom11030467_

Round 1

Reviewer 1 Report

In the present study, Authors searched for novel senolytic compounds through the use of chemoinformatic tools over different chemical databases. The results show that the compounds Hinokitiol, Preussomerin C, and Tanshinone-I can be used as senolytics due to their multitarget potential and their chemical characteristics of similarity. The manuscript is clearly presented and the subject is suitable for Biomolecules, however I have one reflection. Further experimental analysis must be performed to validate senoylitic activity of mentioned compounds, therefore the manuscript should be publish as short communication.

Author Response

Dear reviewer 1: 

Point 1: We appreciate the comments and recommendations on our manuscript; however, we do not consider it feasible for our research article to be re-submitted in a short-communication format again because our study was complex, and this new format would detract from our value and relevance of the research because many of the details within the methodology, obtained results and discussion would be omitted.

Reviewer 2 Report

Comments for manuscript Biomolecules-1142922 “Chemoinformatic screening for the selection of potential senolytic compounds from natural products”

The authors identified three senolytic compounds, such as Hinokitiol, Preussomerin C, and Tanshinone I using chemoinformatics methods. However, experimental assays are required to evaluate the senolytic potential and safety profiles of these compounds. The authors are also indicating the resources of Hinokitiol, Preussomerin C, and Tanshinone I (Table 3). The manuscript is well written in general. Following are some suggestions for improvement:

  1. Please improve the figures 2, 3, and 6 quality.
  2. Relevant references should be added in Tables 1, 2, and 3 (Reference to the pharmacological activity is missing).

Author Response

Dear reviewer 2:

We appreciate the observations given by the reviewer, so in this new version of the manuscript we consider them, and we clarify that the novelty of this study is that no one has previously done a screening within natural products (NPs) to search for  senolytic candidates, in addition to using a novel methodology using the cluster and rcdk packages of the R software when there is no information available such as the QSAR (https://doi.org/10.1186/s13321-019-0405-0), offering this novel methodology as a new alternative when these types of deficiencies exist, in addition to being of great help for the identification of NPs similar to senolytics through their fingerprints. Besides, in this new manuscript, we emphasize that this work aims to avoid unnecessary expenses of time, money, and effort, having as support the use of computational tools, however, it DOES NOT SUBSTITUTE experiments in a laboratory, so as a perspective we include within our paper, the evaluation of our senolytic candidates in an experimental way. Given the rethinking and suggested modifications, we consider that this new version should be considered for publication in your prestigious journal.

1.-Please improve the figures 2, 3, and 6 quality.

Point 1: We appreciate your comments, the quality of these figures already improved in the new version of the manuscript.

2.- Relevant references should be added in Tables 1, 2, and 3 (Reference to the pharmacological activity is missing).

Point 2: We appreciate your comments, references about the pharmacological activity of the compounds were included in the tables of the new version of the manuscript.

Reviewer 3 Report

Biomolecules-1142922

A Brief Summary

The work content is very rich and suitable to publish in Biomolecules in natural product drug discovery. The most important question is the literature survey. The compounds are not sufficient for this study based on PubMed. Please see the attached file for more bioactive senolytic compounds from Scifinder. Also, some minor corrections are required.

Minor revision 

Abstract

  1. Line 24 and 25: Delete the sentence (To our ......Compounds). Please mention the prospect of the senolytic compounds in the medicine.

Introduction:

Line 54: “Idiopathic”  should start with a small letter.

Line 60: Quercetin.: chemical name in the middle of the text should start with a small letter. Please follow the rule for the entire manuscript.

Materials and Methods

2.1.1. Collection of senolytic compounds:

The databases are not sufficient to find out more active senolytic compounds. Scifinder is more reliable. Please collect compounds from scifinder. See attached file

Line 82-83, 146-148: Chemical name start with the small case

Results

Line 167: only 79 compounds are not sufficient for this study.

There is a problem inside the figures:  font size is tiny and not able to legible. Font should increase at least 12 fonts inside all the figures.

Table 1: Please mention only Pubchem CID. The structure should replace with the supplementary file.

Table 2: Please mention specific references for each activity in a separate column.

Author Response

Dear reviewer 3:

We appreciate your commentary.  The senolytics field is relatively recent (it has not more than three decades). As stated in the Methods section, most of these molecules were obtained through an extensive review of all reported senolytic compounds up to September 2020. It is worth mentioning that search recommendation was followed in SciFinder, however despite the site provides a total of 423 results, some of them are reviews and other original papers evaluating the same compound (i.e. quercetin, dasatinib, rapamycin, resveratrol, metformin, and navitoclax) in different models (in vivo and in vitro). Interestingly, the 79 senolytic molecules used in this study are shared on both Pubmed and Scifinder. In further studies, we will consider other specialized search engines.

Minor revision 

 Abstract

Line 24 and 25: Delete the sentence (To our ......Compounds). Please mention the prospect of the senolytic compounds in the medicine.

 Point 1 : We appreciate your comments; this mention has already been included in the new version of the manuscript.

Introduction:

Line 54: “Idiopathic”  should start with a small letter.

 Point 2 : Thanks for the suggestion, we already modify this letter.

Line 60: Quercetin.: chemical name in the middle of the text should start with a small letter. Please follow the rule for the entire manuscript.

 Point 3: We appreciate your suggestion; the small letters on each chemical name in the middle of the text have already been included in the new version of the manuscript.

Materials and Methods

2.1.1. Collection of senolytic compounds:

The databases are not sufficient to find out more active senolytic compounds. Scifinder is more reliable. Please collect compounds from SciFinder. See attached file

Point 4: We appreciate your commentary.  The senolytics field is relatively recent (it has not more than three decades). As stated in the Methods section, most of these molecules were obtained through an extensive review of all reported senolytic compounds up to September 2020. It is worth mentioning that search recommendation was followed in SciFinder, however despite the site provides a total of 423 results, some of them are reviews and other original papers evaluating the same compound (i.e. quercetin, dasatinib, rapamycin, resveratrol, metformin, and navitoclax) in different models (in vivo and in vitro). Interestingly, the 79 senolytic molecules used in this study are shared on both Pubmed and Scifinder. In further studies, we will consider other specialized search engines.

Line 82-83, 146-148: Chemical name start with the small case

Point 5: We appreciate your comment; the cases have already been corrected in line 82-83, 146-148

Results

Line 167: only 79 compounds are not sufficient for this study.

Point 6: We appreciate your commentary, and we agree with you. But, since the senolytics field is relatively new, few molecules have reported this activity at the date. We hope that our study may contribute to enrich and encourage the discovery of more senolytic molecules.

There is a problem inside the figures:  font size is tiny and not able to legible. Font should increase at least 12 fonts inside all the figures.

Point 7: We appreciate your comments and increased the size of the font in this new version of the manuscript.

Table 1: Please mention only Pubchem CID. The structure should replace with the supplementary file.

Point 8: We appreciate the recommendation; the table has already been modified, the structure of each compound has already been replaced by its respective Pubchem CID in this new version.

Table 2: Please mention specific references for each activity in a separate column.

Point 9: We appreciate your comments, and the references about the pharmacological activity of the compounds were included in Table 2 of the new version of the manuscript.
